# SERS Liquid Biopsy Profiling of Serum for the Diagnosis of Kidney Cancer

**DOI:** 10.3390/biomedicines10020233

**Published:** 2022-01-22

**Authors:** Tudor Moisoiu, Stefania D. Iancu, Dan Burghelea, Mihnea P. Dragomir, Gheorghita Iacob, Andrei Stefancu, Ramona G. Cozan, Oana Antal, Zoltán Bálint, Valentin Muntean, Radu I. Badea, Emilia Licarete, Nicolae Leopold, Florin I. Elec

**Affiliations:** 1Clinical Institute of Urology and Renal Transplantation, Clinicilor 2, 400006 Cluj-Napoca, Romania; tudor.moisoiu@umfcluj.ro (T.M.); dan.burghelea@umfcluj.ro (D.B.); media@renaltransplant.ro (G.I.); antal.oanna@gmail.com (O.A.); 2Faculty of Medicine, Iuliu Hațieganu University of Medicine and Pharmacy, Victor Babeș 8, 400012 Cluj-Napoca, Romania; vmuntean@umfcluj.ro (V.M.); rbadea@umfcluj.ro (R.I.B.); 3Biomed Data Analytics SRL, Virgil Onițiu 17, 400696 Cluj-Napoca, Romania; 4Faculty of Physics, Babeș-Bolyai University, Kogalniceanu 1, 400084 Cluj-Napoca, Romania; nicolae.leopold@ubbcluj.ro (N.L.); stefania.iancu@ubbcluj.ro (S.D.I.); andrei.stefancu@ubbcluj.ro (A.S.); ramona.cozan@stud.ubbcluj.ro (R.G.C.); zoltan.balint@ubbcluj.ro (Z.B.); 5Institute of Pathology, Charité-Universitätsmedizin Berlin, Corporate Member of Frei Universität Berlin, Humboldt-Universität zu Berlin and Berlin Institute of Health, Charitépl. 1, 10117 Berlin, Germany; mihnea.dragomir@charite.de; 6Berlin Institute of Health, Anna-Louisa-Karsch-Straße 2, 10178 Berlin, Germany; 7German Cancer Consortium (DKTK), Partner Site Berlin, German Cancer Research Center (DKFZ), 69210 Heidelberg, Germany; 8Octavian Fodor Regional Institute of Gastroenterology and Hepatology, Croitorilor 19-21, 400162 Cluj-Napoca, Romania; 9Faculty of Biology, Babeș-Bolyai University, Republicii 44, 400015 Cluj-Napoca, Romania; emilia.licarete@ubbcluj.ro

**Keywords:** renal cell carcinoma, Raman spectroscopy, SERS, liquid biopsy, machine learning

## Abstract

Renal cancer (RC) represents 3% of all cancers, with a 2% annual increase in incidence worldwide, opening the discussion about the need for screening. However, no established screening tool currently exists for RC. To tackle this issue, we assessed surface-enhanced Raman scattering (SERS) profiling of serum as a liquid biopsy strategy to detect renal cell carcinoma (RCC), the most prevalent histologic subtype of RC. Thus, serum samples were collected from 23 patients with RCC and 27 controls (CTRL) presenting with a benign urological pathology such as lithiasis or benign prostatic hypertrophy. SERS profiling of deproteinized serum yielded SERS band spectra attributed mainly to purine metabolites, which exhibited higher intensities in the RCC group, and Raman bands of carotenoids, which exhibited lower intensities in the RCC group. Principal component analysis (PCA) of the SERS spectra showed a tendency for the unsupervised clustering of the two groups. Next, three machine learning algorithms (random forest, kNN, naïve Bayes) were implemented as supervised classification algorithms for achieving discrimination between the RCC and CTRL groups, yielding an AUC of 0.78 for random forest, 0.78 for kNN, and 0.76 for naïve Bayes (average AUC 0.77 ± 0.01). The present study highlights the potential of SERS liquid biopsy as a diagnostic and screening strategy for RCC. Further studies involving large cohorts and other urologic malignancies as controls are needed to validate the proposed SERS approach.

## 1. Introduction

Renal cancer (RC) represents 3% of all cancers, with a 2% annual incidence increase worldwide [1,2]. Renal cell carcinoma (RCC) is the most prevalent histologic subtype of RC, accounting for approximately 90% of all renal malignancies [3]. In recent years, the survival outcomes of RCC have improved significantly. Nonetheless, up to a third of patients are diagnosed with regional or distant metastasis [4,5], with a 5-year overall survival rate of around 12% for metastatic cases, thus opening the discussion about the need for screening and developing new diagnostic tools that can facilitate early detection [5,6]. Several screening strategies have been considered for RCC, but none of them are currently approved. For instance, despite having good sensitivity and specificity, contrast-enhanced computer tomography (CECT) is not suitable for this purpose because of its invasive nature, high irradiation, and lack of cost effectiveness [7]. Ultrasonography has emerged as a potential screening tool, but the accuracy is operator dependent and influenced by tumor size and location. For example, Jamis-Dow et al. reported an accuracy of just 62% for tumors less than 3 cm [8]. Several serum and urine biomarkers have also been studied, but none of them have yet been validated [9,10,11].

In recent years, surface-enhanced Raman scattering (SERS) profiling of biofluids has shown promising results as a screening and diagnostic method, especially with the development of low-cost and easy-to-use Raman spectroscopes [12].

Raman spectroscopy is a type of vibrational spectroscopy based on the inelastic scattering of laser photons, which provides information concerning the molecular structure of samples [13]. The main advantages of Raman spectroscopy are the ability to analyze samples with little to no pre-processing steps, having a fast turnaround time, and being amenable to implementation in point-of-care settings. However, the use of Raman spectroscopy for biomedical applications is hindered by the relatively low sensitivity for low concentrations of analytes such as those from serum or urine. SERS is a method to amplify the Raman signal of analytes based on the adsorption on metallic nanostructures [12]. Since only the Raman signal of molecules adsorbed on the SERS substrate is amplified, SERS can greatly improve the specific detection of analytes from complex biological fluids, especially purine metabolites.

Our group previously demonstrated the possibility of using SERS profiling of biofluids to detect several types of malignancies, including breast, gastrointestinal, lung, ovarian, oral, and prostate cancer [14,15,16,17]. However, no such study has yet been performed in the case of RCC. In this pilot study, the diagnostic accuracy of SERS profiling of serum for RCC was explored for the first time, with the aim of highlighting its potential use as a screening tool.

## 2. Materials and Methods

### 2.1. Patients

We prospectively enrolled 23 patients with RCC (10 patients with Stage 1 RCC, 2 patients with Stage 2, and 11 patients with Stage 3) attending the Clinical Institute of Urology and Renal Transplantation, Cluj-Napoca, Romania. In parallel, we enrolled 27 controls (CTRL) presenting with other non-malignant pathologies (Appendix A). The pathology assessment for the RCC group was conducted using the AJCC 7th edition TNM classification [18]. The study was approved by the Ethics Committee of the Clinical Institute of Urology and Renal Transplantation (Document No. 1/2018).

### 2.2. Sample Collection

An amount of 10 mL of blood was collected in serum separator tubes from patients before any treatment. Serum was separated by centrifugation at 425 g for 5 min and subsequently stored at −80 °C until further analysis.

### 2.3. SERS Profiling

For the SERS analysis, 50 µL of serum was mixed with 450 µL of methanol and centrifuged for 10 min at 5800 g. The supernatant was carefully collected for further analysis. Silver nanoparticles synthesized by reduction with hydroxylamine hydrochloride (hya-AgNPs) were used as SERS substrates [19]. The SERS analysis was performed on a mixture of 45 µL of hya-AgNPs activated with Ca^2+^ (final concentration of Ca(NO_3_)_2_ was 10^−4^ M) and 5 µL of serum. A drop of 5 µL from this mixture was deposited on a microscope slide covered with aluminum foil, and the SERS spectra were immediately acquired. SERS spectra were acquired using an InVia Raman Spectrometer (Renishaw) equipped with a 532 nm laser (180 mW) coupled to an upright Leica microscope. The 532 nm laser was focused on the sample through a 5X objective (Leica, NA = 0.12). Each measurement consisted of an average of 3 acquisitions, 20 s of integration each time. The laser power on the sample was set to 10% (18 mW).

### 2.4. Statistical Analysis

The raw SERS spectra were pre-processed by selecting only the spectral region between 400 and 1800 cm^−1^ for further analysis. The subsequent pre-processing steps included vector normalization, rubber-band baseline subtraction, and smoothing (Savitzky–Golay, with the window set to 5 and the polynomial order set to 2).

Next, principal component analysis (PCA) was performed to reduce the dataset’s dimensionality and allow the visualization of the unsupervised clustering of the RCC and CTRL groups. To select relevant principal components (PCs) that allowed for discrimination between the RCC and CTRL groups, Student’s *t*-test was employed. A probability *p*-value of less than 0.05 was considered significant.

The selected PCs were then used as inputs for three machine learning algorithms (random forest, kNN, and naïve Bayes) that were trained to discriminate between the RCC and CTRL groups. The machine learning algorithms were internally validated using leave-one-out cross-validation.

The statistical analysis was performed using Quasar-Orange software, Orange-Spectroscopy library (Bioinformatics Laboratory of the University of Ljubljana) [20].

## 3. Results

The average SERS spectra of the RCC and CTRL groups, together with their standard deviation, are shown in Figure 1A.

The SERS spectra of serum were dominated by the SERS bands attributed to purine metabolites (uric acid, hypoxanthine, and xanthine) and carotenoids (Table 1). Although the detection of carotenoids is based on resonant Raman scattering and not SERS, their resonant Raman signal appears in the recorded biofluid SERS spectrum [21].

The major differences between the SERS spectra of the RCC and CTRL groups are represented by the carotenoids’ bands (1155 and 1520 cm^−1^), which were less intense in the RCC group. In contrast, SERS bands of purine metabolites (534, 590, 638, 725, 811, 890, 1130, 1357, 1450, 1560, and 1684 cm^−1^) showed higher intensities in the RCC group.

The first nine PCs were kept for further analysis, which explained 98% of the variance in the initial dataset (Appendix A). The differences in score values of the nine PCs between the RCC and CTRL groups were tested by Student’s *t*-test. The scores of PC2 and PC6 showed statistically significant differences between the two groups (*p* < 0.05) (Figure 1B). The score plot for PC2 and PC6 showed a clear tendency of clustering of the two groups (Figure 1C). The loading plots corresponding to PC2 and PC6 are shown in Figure 1D, highlighting the main contributors to the clustering of the SERS spectra. Thus, PC2 was dominated by the carotenoid bands at 1155 and 1520 cm^−1^, while PC6 was dominated by bands attributed to purine metabolites.

To test the performance of SERS profiling as a screening tool for RCC, three machine learning algorithms were used (random forest, kNN, and naïve Bayes). The two previously selected PCs were employed as input for the machine learning algorithms, yielding an AUC of 0.78 for random forest, 0.78 for kNN, and 0.76 for naïve Bayes (average AUC 0.77 ± 0.01) (Figure 2). The performance metrics of the three classifiers after internal leave- one-out cross-validation are presented in Table 2.

To explore the effect of cancer stage on classification accuracy, the three machine learning algorithms were trained to discriminate between the CTRL group, Stage 1 RCC subgroup, and Stage 3 RCC subgroup. For each comparison, the most significant PCs were used (Supplementary Appendix A). As expected, the diagnostic accuracy in discriminating high Stage 3 RCC was slightly better (average AUC 0.79 ± 0.08) than in the case of Stage 1 RCC (average AUC 0.72 ± 0.05) (Appendix A). No Stage 2 RCC subgroup analysis was performed because of the small number of patients (*n* = 2). The average classification accuracy yielded by the three machine learning algorithms in discriminating between Stage 1 RCC and Stage 3 RCC was 0.80 ± 0.1 (Appendix A).

A lower intensity of the resonant Raman bands at 1155 cm^−1^ and 1520 cm^−1^ attributed to carotenoids was also previously reported in serum samples of breast, gastrointestinal, lung, ovarian, oral, and prostate cancer patients [15].

## 4. Discussion

The increasing annual number of newly discovered RCC cases and the high number of locally advanced or metastatic stages have prompted a quest for new efficient screening strategies. In this study, we demonstrated the possibility of discriminating between the RCC and CTRL groups based on the SERS spectra of deproteinized serum. From a clinical perspective, SERS-based profiling could be used as a preliminary screening strategy, followed by detailed imaging studies with CECT, contrast-enhanced MRI, or contrast-enhanced ultrasound in patients in whom malignancy is suggested by SERS profiling.

The results suggest that the ability of SERS profiling of serum to discriminate RCC patients stems from the fact that they exhibit higher levels of purine metabolites (uric acid, xanthine, and hypoxanthine), but lower levels of carotenoids (Figure 1 and Table 1).

Hypoxanthine, xanthine, and uric acid are the last components in the catabolism of purine nucleotides adenosine and guanosine phosphate. Extensive epidemiological evidence showed that elevated baseline serum uric acid (hyperuricemia) is associated with increased cancer risk across cancer types [26], and in RCC in particular [27]. The effect seems to be exerted by the upregulation of key components of chronic inflammatory pathways such as adiponectin, C-reactive protein, and leptin [26]. Hyperuricemia is caused by the imbalance between uric acid production and excretion because of a multitude of genetic and environmental factors, including diet and alcohol consumption [26]. In addition, once the malignant process has started, uric acid is continuously released from dead and dying cells. The higher intensity in SERS bands attributed to uric acid in the RCC group is in line with previous SERS studies concerned with breast, gastrointestinal, lung, ovarian, oral, and prostate cancer.

Besides uric acid, a higher intensity of the SERS bands attributed to hypoxanthine (725, 1450, and 1684 cm^−1^) and xanthine (1684 cm^−1^) was also noted in the case of the RCC group, in accordance with previous studies concerned with breast, gastrointestinal, lung, ovarian, oral, and prostate cancer [15]. Increased levels of hypoxanthine in patients with RCC were previously reported in a metabolomic study by Monteiro et al. [28], with the authors also reporting increased levels of this metabolite in older and smoking patients, an effect that possibly confounds hypoxanthine as an RCC marker. Mechanistically, the increase in the levels of hypoxanthine and xanthine might be the result of the cancer-associated downregulation of xanthine oxidoreductase (XOR), the enzyme that catalyzes the conversion of xanthine and hypoxanthine to uric acid [29]. However, this mechanism is not specific to RCC.

In contrast to the case of the SERS bands attributed to purine metabolites, which exhibited low intensities in RCC, the resonant Raman bands at 1155 cm^−1^ and 1520 cm^−1^ attributed to carotenoids exhibited lower intensities (Figure 1A and Table 1). Carotenoids are a class of more than 750 naturally occurring organic pigments [30]. The most studied carotenoids are beta-carotene, lycopene, lutein, and zeaxanthin, which are found in various fruits and vegetables. Low dietary intake or blood concentrations of carotenoids have been linked to a higher incidence of cardiovascular diseases, cancer, and all-cause mortality [31]. A similar effect was also noted for RCC [32]. Carotenoids inhibit oxidative damage to DNA, mutagenesis, tumor growth, and malignant transformation, and enhance cell–cell communication, thereby protecting cells against malignant transformation [32]. However, similar to purine metabolites, perturbations in carotenoids are not specific to RC.

The PCA of the SERS spectra identified two PCs that were significantly different between the RCC and CTRL groups (PC2 and PC6). PC2 was dominated by negative bands at 1155 cm^−1^ and 1520 cm^−1^ attributed to carotenoids, and the SERS spectra pertaining to the RCC group clustered towards the negative side of the axis, in line with the lower intensity of these SERS bands in the RCC group. Conversely, PC6 was dominated by positive SERS bands attributed to purine metabolites, and as expected, spectra from the RCC group were positioned towards higher PC6 score values. Taken together, the results of the PCA suggest that the spectral information concerning purine metabolites and carotenoids was sufficient to achieve a good separation between the RCC and CTRL groups.

To quantify the classification accuracy yielded by SERS profiling, three machine learning algorithms were trained to discriminate between the RCC and CTRL groups, yielding an AUC of 0.78 for random forest, 0.78 for kNN, and 0.76 for naïve Bayes (average AUC 0.77 ± 0.01) (Table 2)).

While no previous study has been reported yet concerning SERS profiling of biofluids in RCC patients, there are several studies employing Raman spectroscopy for the analysis of pathology specimens (see meta-analysis by Jin et al. [33]), as well as one study involving the SERS analysis of homogenized tissue samples [30]. Thus, SERS profiling of homogenized tissue samples yielded an overall accuracy of 0.93, higher than the overall accuracy of 0.77 reported in the present study (Table 2). It is important to note that when analyzing homogenized tissue by SERS, the types of molecules amenable to detection are broader and include proteins. On the other hand, since the method requires excised tissue samples, it cannot be implemented as a liquid biopsy or screening strategy, representing an important limitation. 

While perturbations in purine metabolites and carotenoids are a general feature of malignancies, making purine metabolites and carotenoids non-specific markers, we have previously demonstrated the possibility of attaining a differential diagnosis between breast, colorectal, lung, ovarian, and oral cancer based on subtle differences in the SERS spectra of serum [15]. Whether this feat can also be achieved in the case of RCC is currently unknown.

Other liquid biopsy strategies have also been considered for RCC, the most advanced marker for its detection being kidney injury molecule-1 (KIM-1), which yielded an accuracy of around 75% [34]. Given the relatively low incidence of RC, the clinical implementation of a screening strategy with such accuracy would translate to many healthy subjects requiring invasive procedures to rule out false positive findings. MicroRNA profiling has also been explored for RCC diagnosis and screening, with a recent review by Sequira et al. reporting accuracies ranging from 0.7 to 0.93 for various miRNA panels [11]. Taken together, the current landscape of liquid biopsy strategies for RCC detection is promising. Still, there is a lack of markers validated in large, prospective, randomized studies [9], highlighting the need for novel approaches tackling this issue.

One of this study’s main limitations is represented by the small number of enrolled patients. Moreover, the control groups should also have ideally included patients with other types of urologic malignancies (e.g., prostate and bladder cancer), since all these malignancies share the same clinical presentation and subsequent work-up. Ideally, the study should have included an external validation cohort, as recommended by the REMARK guideline on study methodology for new tumor marker studies [35]. Instead, internal leave-one-out cross-validation was used to validate the model because an external cohort was not available, as recommended by the same authors [35]. The inclusion of an external validation cohort is mandatory in future studies. As an additional limitation, the SERS spectra were acquired using a state-of-the-art Raman spectroscope operated under the ideal setting of a laboratory. We recently reported that the diagnosis of gastrointestinal cancer using a portable Raman spectroscope operated in a clinical setting is feasible [14]. Whether the use of a more realistic scenario for the acquisition of SERS spectra would have impacted the accuracy is unknown.

Another issue that we did not address in this study concerns the evaluation of renal cysts, which are vesicular lesions that can be either benign or malignant. The work-up for renal cysts requires contrast-enhanced computer tomography, which allows the classification of the cysts using the Bosniak scale. In the case of Bosniak III cysts, which are defined as indeterminate cystic masses with thickened irregular walls or septa with enhancement and which have an approximately 50% chance of being malignant [9], partial or radical nephrectomy is warranted. Existing non-invasive methods of reducing the number of unnecessary nephrectomies are suboptimal, as highlighted by a recent study evaluating contrast-enhanced ultrasonography (CEUS), which reported that only 63% of CEUS-diagnosed Bosniak III cysts were malignant after partial or radical nephrectomy [36]. Whether SERS is also able to differentiate between benign and malignant cystic lesions is currently unknown and will require future studies.

## 5. Conclusion

In this study, we demonstrated, for the first time, that SERS profiling of serum allows for the detection of RCC by capturing key perturbations in purine and carotenoid metabolism, yielding an average accuracy of 0.77 based on three different machine learning algorithms. Future studies assessing the accuracy of SERS profiling in large prospective cohorts with external validation are warranted to validate this strategy and translate it to the current clinical setting.

## Figures and Tables

**Figure 1 biomedicines-10-00233-f001:**
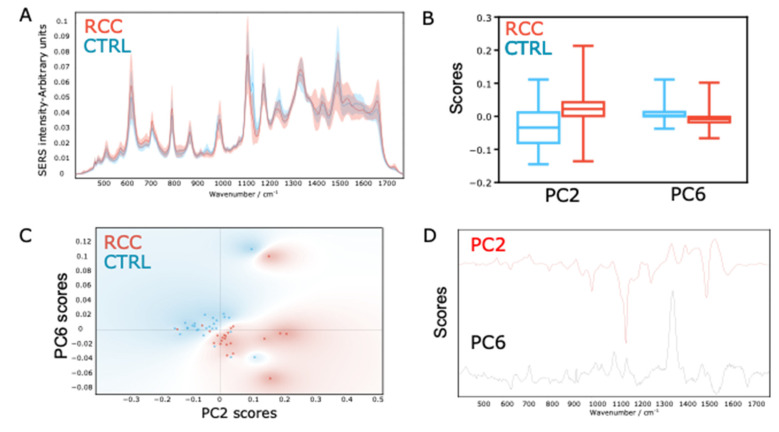
(**A**) The average SERS spectra of serum from renal cell carcinoma (RCC) versus control (CTRL) patients. (**B**) The distribution of score values for principal component (PC) 2 and PC6 of RCC (red) and CTRL (blue) patients. (**C**) Score plots of PC2 and PC6 for RCC (red) and CTRL (blue) patients. (**D**) Loading plots of PC2 and PC6.

**Figure 2 biomedicines-10-00233-f002:**
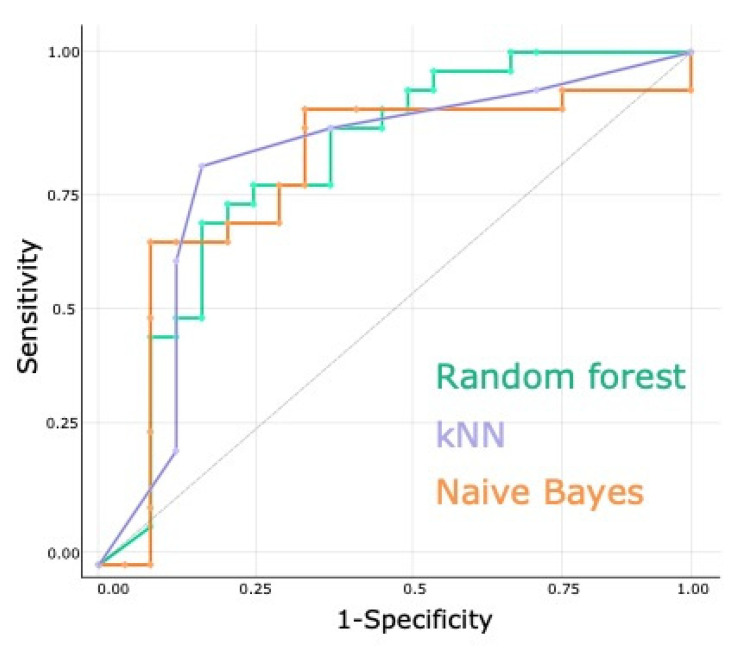
Head-to-head comparison of the receiver operating characteristic (ROC) curves for the classification accuracy yielded by SERS analysis of serum from renal cancer and control patients using three supervised classification algorithms (random forest, kNN, naïve Bayes).

**Table 1 biomedicines-10-00233-t001:** Tentative assignment of the SERS bands [22,23,24,25].

Metabolite	SERS Band Assignment (cm^−1^)
Uric acid	534, 590, 638, 811, 890, 1130, 1204, 1260, 1357, 1560, 1684
Hypoxanthine	725, 1450, 1684
Xanthine	1357, 1684
Carotenoids	1155, 1520
Methanol	1015

**Table 2 biomedicines-10-00233-t002:** The performance metrics for the classification of renal cell carcinoma and control group patients based on surface-enhanced Raman spectroscopy (SERS) spectra of serum using three classification algorithms (random forest, kNN, and naïve Bayes). AUC—area under the curve; CA— classification accuracy; F1—score represents the harmonic mean of precision and recall; Precision—positive predicted values; Recall—sensitivity.

Machine Learning Model	AUC	CA	F1	Precision	Recall
Random forest	0.78	0.72	0.71	0.72	0.72
kNN	0.78	0.80	0.80	0.80	0.80
Naïve Bayes	0.76	0.70	0.69	0.69	0.70

## Data Availability

Not applicable.

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
