# Peer review of "SERS Liquid Biopsy Profiling of Serum for the Diagnosis of Kidney Cancer"

_biomedicines, 2022, doi:10.3390/biomedicines10020233_

Round 1

Reviewer 1 Report

The authors conducted a SERS liquid biopsy profiling on samples were collected from 23 patients with renal cell carcinoma and 27 controls. The authors carried out three different machine learning algorithms, and they were cross validated using repeated random subsampling validation. They concluded that key perturbations in purine and carotenoid metabolism yielded a high accuracy of differential diagnosis. Although the reviewer agrees that the findings are of interest and contain novelty, there are several issues in this study that should be addressed before considered for publication.

Major points

  1. What are main points of this study from clinical perspectives? Admitting that it is useful to predict the presence of renal cell carcinoma, which does not have specific tumour marker, from peripheral blood, the reviewer believes that the diagnosis should be based on (contrast) computed tomography. Does SERS surpass current imaging device?
  2. The reviewer is concerned about reproducibility of the results. The authors only conducted internal validation using 20% of the full cohort, which is relatively small. Lack of external validation is a great drawback of this study and there is a risk of overfitting from it.
  3. The findings from SERS profiling should correspond with biological aetiology of renal cell carcinoma. What roles do purine metabolites and carotenoids play in the carcinogenesis of renal cell carcinoma? These should be discussed in more detail.

Minor points

  1. There is a typography in Line 125 ‘more less’.

Author Response

We thank the reviewers for their insightful suggestions. Please find below our point-by-point responses. The changes in the manuscript are highlighted by the track-change option of MS Word. We feel that our manuscript has now improved significantly with these comments, and we hope you find it suitable for publication in Biomedicines journal.

Reviewer 1

Major points

  1. What are the main points of this study from clinical perspectives? Admitting that it is useful to predict the presence of renal cell carcinoma, which does not have a specific tumor marker, from peripheral blood, the reviewer believes that the diagnosis should be based on (contrast) computed tomography. Does SERS surpass current imaging device?

Response: We thank the reviewer for this observation. The main point from the clinical perspective is that SERS-based profiling could be used only as a non-invasive screening strategy and that the diagnosis should be based on contrast-enhanced CT (CECT) in patients in which renal cancer is suggested by the results of the SERS profiling, in line with the reviewer’s comment. In fact, CECT is also important in the staging and treatment planning of renal masses, making it indispensable in the workup of renal cancer patients.

Regarding the accuracy of imaging techniques, a recent metanalysis reported an AUC of 0.73 for CECT, an AUC of 0.75 for contrast-enhanced MRI and an AUC of 0.78 for contrast-enhanced ultrasound in differentiating between benign and malignant renal masses (PMID: 31570270), which is comparable to the AUC reported in this study (0.77). However, comparing the accuracy of SERS with that of standard imaging techniques is difficult since only limited data exists regarding SERS. In addition, the clinical scenarios in which SERS profiling and imaging are performed are different: imaging techniques are used in patients presenting with specific symptoms or in which a renal mass has been found incidentally, while SERS is intended as a screening strategy in the general population, making a direct comparison of accuracy difficult.

Following the reviewer’s suggestion, in the revised manuscript, the following paragraph was added (line 345):

” From a clinical perspective, SERS-based profiling could be used as a preliminary screening strategy, followed by detailed imaging studies with CECT, contrast-enhanced MRI or contrast-enhanced ultrasound in patients in which malignancy is suggested by SERS profiling.”

  1. The reviewer is concerned about the reproducibility of the results. The authors only conducted internal validation using 20% of the full cohort, which is relatively small. Lack of external validation is a great drawback of this study, and there is a risk of overfitting from it.

Response: We agree with the reviewer that in small cohorts such as these, there is a risk of overfitting. Consequently, while a 5-fold cross-validation is an acceptable strategy (i.e., using 80% of the cohort for training and 20% of the cohort for validation), to minimize the risk of overfitting, in the revised manuscript we employed a more stringent cross-validation strategy (namely leave-one-out cross-validation), which yielded an AUC of 0.77 ± 0.01.

  1. The findings from SERS profiling should correspond with biological etiology of renal cell carcinoma. What roles do purine metabolites and carotenoids play in the carcinogenesis of renal cell carcinoma? These should be discussed in more detail.

Response: The association between purine metabolites and carotenoids and renal cancer is based on observational studies and mechanistic insights are scarce (see also the response to the 2nd comment of Reviewer 2 regarding the specificity of the findings for renal cancer). Nonetheless, one might argue that the empirical observations are in line with the fact that renal cancer is a metabolically driven disease, as many known genes associated with the development of renal cancer are involved in regulating cellular metabolism within nutrient-deprived tumor microenvironments (PMID: 28480903).

In the revised manuscript, we added a paragraph concerned with the molecular mechanisms associated with cancer protecting effects of carotenoids and we highlighted the fact that the perturbations in purine metabolites and carotenoids are not specific to renal cancer.

Following the reviewer’s suggestion, in the revised manuscript, the following paragraph was added (line 421):

“Carotenoids inhibit oxidative damage to DNA, mutagenesis, tumor growth, malignant transformation, and enhance cell-cell communication, thereby protecting cells against malignant transformation [32]. However, similarly to purine metabolites, perturbations in carotenoids are not specific to RC”.

Minor points 

  1. There is typography in Line 125 ‘more less’.

Response: We have corrected the typography, thank you.

Reviewer 2 Report

Dear Authors, 

this is an interesting work and nicely designed but several points have to be adressed:

  • in my opinion title has to changed. Pont of care suggests easy to use screening tool for RCC. Results of your work do not prove it.
  • as you mentioned in discussion purine metabolism alterations are not RCC specific. They happen in lots of malignances. Because of that your results can not be interpreted as potential screening tool for RCC, but for malignancies in general RCC included
  • so far screening for RCC is not beeing pursued because this malignancy is not often enough in population. Compare to breast, lung or prostate. Even in prostate (most common malignancy in men) screening is questionable. Additionally as you mentioned screening tools (CT or ultrasound) are not satisfactory for various reasons. Compare your method to PSA testing, this type of testing is maximum what a patient and health system can bear. Your method on this stage of deveopment can not be used and office based testing. Please consider those arguments before making statements regarding using Raman spectroscopy as screenieng tool
  • please consider comparison in your small group between RCC group of low disease burden (T1), moderate (T2) and high (T3-4) between groups and healthy volunteers. Results might be interesting and might add a lot to your manuscript.  

Author Response

We thank the reviewers for their insightful suggestions. Please find below our point-by-point responses. The changes in the manuscript are highlighted by the track-change option of MS Word. We feel that our manuscript has now improved significantly with these comments, and we hope you find it suitable for publication in Biomedicines journal.

Reviewer 2

  1. In my opinion title has to changed. Point-of-care suggests easy to use screening tool for RCC. Results of your work do not prove it.

Response: In line with the reviewer’s suggestion, we removed” point-of-care” from the title.

  1. As you mentioned in discussion purine metabolism alterations are not RCC specific. They happen in lots of malignancies. Because of that your results can not be interpreted as a potential screening tool for RCC, but for malignancies in general RCC included.

Response: While perturbations in purine metabolites and carotenoids are a general feature of malignancies, making purine metabolites and carotenoids non-specific markers, we have previously demonstrated the possibility to attain a differential diagnosis between breast, colorectal, lung ovarian, and oral cancer based on subtle differences in the SERS spectra of serum [15]. Whether this feat can also be achieved in the case of renal cancer is currently unknown.

Following the reviewer’s suggestion, in the revised manuscript, the following paragraph was added (line 338):

“A lower intensity of the resonant Raman bands at 1155 cm-1 and 1520 cm-1 attributed to carotenoids was also previously reported in serum samples of breast, gastrointestinal, lung, ovarian, oral, and prostate cancer patients [15]”.

  1. So far screening for RCC is not being pursued because this malignancy is not often enough in population.

Response: While the incidence of RCC is relatively low, the continuous increase in its incidence has led to many experts demanding the development and implementation of screening strategies [5-6].

  1. Compare to breast, lung or prostate. Even in prostate (most common malignancy in men) screening is questionable. Additionally as you mentioned screening tools (CT or ultrasound) are not satisfactory for various reasons. Compare your method to PSA testing, this type of testing is maximum what a patient and health system can bear.

Response: The reason why screening for prostate cancer based on PSA is questionable is because of the low accuracy of the method. Thus, the estimated AUC of PSA is around 0.66 (PMID: 25403590). Consequently, accuracy is the main barrier in the clinical implementation of screening strategies, and any screening strategy which is accurate enough has the potential for clinical translation. However, this condition is even more stringent in the case of cancer types with lower incidence, such as renal cancer.

In the revised manuscript, we added a paragraph stating that given the low incidence of renal cancer, the clinical translation of any screening strategy would require high accuracy in order to prevent the situation of many patients undergoing invasive procedures to rule out false-positive results. In contrast, even screening strategies with lower accuracies are still acceptable in the case of prevalent types of cancer such as prostate and breast cancer.

In the revised manuscript, the following paragraph was added (line 515):

“Given the relatively low incidence of RC, the clinical implementation of a screening strategy with such an accuracy would translate in many healthy subjects requiring invasive procedures to rule out false-positive findings”.

  1. Your method on this stage of development can not be used and office-based testing. Please consider those arguments before making statements regarding using Raman spectroscopy as screening tool

Response: While in the present study the spectra were acquired using a state-of-the-art Raman spectroscope, we have recently reported that a portable Raman spectroscope operating in a clinical scenario similar to office-based testing represents a viable and promising strategy for the clinical implementation of SERS-based screening strategies [14].

In the revised manuscript, the following paragraph was added (line 527):

“As an additional limitation, the SERS spectra were acquired using a state-of-the-art Raman spectroscope operated under the ideal setting of a laboratory. We have recently reported that the diagnosis of gastrointestinal cancer using a portable Raman spectroscope operated in a clinical setting is feasible [14]. Whether the use of a more realistic scenario for the acquisition of SERS spectra would have impacted accuracy is unknown”.

  1. Please consider comparison in your small group between RCC group of low disease burden (T1), moderate (T2) and high (T3-4) between groups and healthy volunteers. Results might be interesting and might add a lot to your manuscript.  

Response: We thank the reviewer for this excellent suggestion. As expected, the classification accuracy for Stage 3 RCC was slightly better than for Stage 1 RCC group (AUC=0.79 vs AUC=0.72) (see Supplementary Table S3 and Supplementary Table S4). In addition, it was possible to differentiate between T1 and T3 RCC groups with high accuracy (Supplementary Table S5). However, we did not add the results to the main manuscript because of the low number of samples in each subgroup (n = 10 for Stage 1 RCC, n= 11 for Stage 3 RCC), which limits the statistical relevance of the stage-related classification. 

Round 2

Reviewer 1 Report

Thanks for the extensive revision. The authors have responded to the reviewer’s comments, there is only one point to be addressed in more detail before considered for publication.

Major point

  1. The reviewer is concerned about the reproducibility of the results. The authors only conducted internal validation using 20% of the full cohort, which is relatively small. Lack of external validation is a great drawback of this study, and there is a risk of overfitting from it.

Response: We agree with the reviewer that in small cohorts such as these, there is a risk of overfitting. Consequently, while a 5-fold cross-validation is an acceptable strategy (i.e., using 80% of the cohort for training and 20% of the cohort for validation), to minimize the risk of overfitting, in the revised manuscript we employed a more stringent cross-validation strategy (namely leave-one-out cross-validation), which yielded an AUC of 0.77 ± 0.01.

>> Admitting that jackknife methods in general possess less bias, the reviewer considers a new model needs to be externally validated using an independent cohort according to the REMARK guideline [Altman DG, et al. PLoS Med. 2012 May; 9(5): e1001216.]. Therefore, it is not sufficient to use a subgroup of the same population given that there is always a risk of selection and information bias.

This point should further be discussed in the ‘Discussion’ and ‘Conclusion’ sections of the manuscript.

Author Response

We thank the reviewers for their suggestions. In our opinion, their suggestions helped to increase the quality of the article. The changes in the manuscript are highlighted by the track-change option of MS Word.

Reviewer 1.

  1. Admitting that jackknife methods in general possess less bias, the reviewer considers a new model needs to be externally validated using an independent cohort according to the REMARK guideline [Altman DG, et al. PLoS Med. 2012 May; 9(5): e1001216.]. Therefore, it is not sufficient to use a subgroup of the same population given that there is always a risk of selection and information bias.

Response: We thank the reviewer for this observation. Indeed, external validation represents the best way for the validation of a new tumor marker, as beautifully explained by Altman et all in the REMARK guideline. Unfortunately, in these troubling times, it was not possible to externally validate our results. For this reason, as suggested, we used internal validation with leave-one-out cross-validation. To better highlight this limitation, we made the next changes:

Line 113: “The machine learning algorithms were internally validated using leave-one-out cross-validation.”

Line 147: “The performance metrics of the three classifiers after internal leave-one-out cross-validation are presented in Table 2.”

Line 261: Ideally, the study should have included an external validation cohort, as recommended by the REMARK guideline on study methodology for new tumor markers studies [35]. Instead, internal leave-one-out cross-validation was used to validate the model because an external cohort was not available, as recommended by the same authors [35]. The inclusion of an external validation cohort is mandatory in future studies”.

Line 288: “Future studies assessing the accuracy of SERS profiling in large prospective cohorts with external validationare warranted to validate this strategy and translate it in the current clinical setting.”

Reviewer 2 Report

Thank you for modifications. pleas follow this interesting research path

Author Response

Thank you for your help! Hopefully, we will continue the research on this topic.